# Improving *Agrobacterium tumefaciens*−Mediated Genetic Transformation for Gene Function Studies and Mutagenesis in Cucumber (*Cucumis sativus* L.)

**DOI:** 10.3390/genes14030601

**Published:** 2023-02-27

**Authors:** Hanqiang Liu, Jianyu Zhao, Feifan Chen, Zhiming Wu, Junyi Tan, Nhien Hao Nguyen, Zhihui Cheng, Yiqun Weng

**Affiliations:** 1Horticulture Department, University of Wisconsin, Madison, WI 53706, USA; 2College of Horticulture, Northwest A&F University, Yangling 712100, China; 3Institute of Cash Crops, Hebei Academy of Agriculture & Forestry Sciences, Shijiazhuang 050051, China; 4USDA-ARS Vegetable Crops Research Unit, Madison, WI 53705, USA

**Keywords:** cucumber, *Cucumis sativus*, genetic transformation, *Agrobacterium tumefaciens*, T−DNA insertion mutagenesis, gene editing

## Abstract

In the post−genomics era, *Agrobacterium tumefaciens*−mediated genetic transformation is becoming an increasingly indispensable tool for characterization of gene functions and crop improvement in cucumber (*Cucumis sativus* L.). However, cucumber transformation efficiency is still low. In this study, we evaluated the effects of several key factors affecting the shoot−regeneration rate and overall transformation efficiency in cucumber including genotypes, the age and sources of explants, *Agrobacterium* strains, infection/co−cultivation conditions, and selective agents. We showed that in general, North China cucumbers exhibited higher shoot−regeneration rate than US pickling or slicing cucumbers. The subapical ground meristematic regions from cotyledons or the hypocotyl had a similar shoot−regeneration efficiency that was also affected by the age of the explants. Transformation with the *Agrobacterium* strain AGL1 yielded a higher frequency of positive transformants than with GV3101. The antibiotic kanamycin was effective in selection against non−transformants or chimeras. Optimization of various factors was exemplified with the development of transgenic plants overexpressing the *LittleLeaf* (*LL*) gene or RNAi of the *APRR2* gene in three cucumber lines. The streamlined protocol was also tested in transgenic studies in three additional genes. The overall transformation efficiency defined by the number of verified transgenic plants out of the number of seeds across multiple experiments was 0.2–1.7%. Screening among T_1_ OE transgenic plants identified novel, inheritable mutants for leaf or fruit color or size/shape, suggesting T−DNA insertion as a potential source of mutagenesis. The *Agrobacterium*−mediated transformation protocol from this study could be used as the baseline for further improvements in cucumber transformation.

## 1. Introduction

Cucumber (*C. sativus* L.) is an economically valuable vegetable crop worldwide and a model of choice to investigate several important biological processes such as chromosome evolution, sex expression, and vascular biology [1]. Since public release of the cucumber draft genomes over a decade ago [2,3], significant progress has been made in molecular mapping and cloning of genes or quantitative trait loci (QTL) in cucumber (reviewed in [4,5,6,7,8,9]). Very few of those cloned candidate genes or QTL have been functionally characterized. Recent advances in the Clustered Regularly Interspaced Short Palindromic Repeats−associated protein 9 (CRISPR−Cas9)−mediated gene editing have revolutionized plant biology research and crop improvement [10]. An efficient genetic transformation system is important for gene function studies or gene editing; however, this is lacking in cucumber. 

Genetic transformation is a process in which a cell takes up naked DNA from the surrounding medium and incorporates it to acquire an altered genotype that is heritable [11]. The Gram−negative bacterium *A. tumefaciens* (*Agrobacterium* in short hereinafter) has the ability to transfer DNA into plant cells and integrate this T−DNA stably into the nuclear genome [12]. By taking advantage of this natural engineer, *Agrobacterium*−mediated transformation has becoming the most widely used genetic transformation approach in both dicot and monocot plants. Trulson et al. [13] were probably the first to report genetic transformation in cucumber. Since then, extensive studies have been conducted to develop and optimize *Agrobacterium*−mediated transformation in cucumber (reviewed in [14,15,16]). A typical *Agrobacterium*−mediated genetic transformation experiment includes the selection and preparation of explants, development of T−DNA plasmid constructs, inoculation and co−cultivation of explants with *Agrobacterium*, shoot regeneration in selective growth media, rooting of shoots and generation of plantlets, and finally the validation and functional characterization of target genes in transgenic plants. Many factors have been shown to affect the transformation efficiency that may include genotypes; explant sources; the strains, concentrations, and inoculation methods of *Agrobacterium;* the temperature, duration, and light conditions of co−cultivation of explants with the bacterium; components and concentrations of chemicals in culture media; selective antibiotics; and reporter genes for positive transformants. Many investigations have been carried out aiming to optimize various parameters to improve the transformation efficiency (reviewed in [4,14,15,16]). Some of these factors seem to be consistent and readily applicable across different experiments. For example, most studies used cotyledon explants cultured in the Murashige and Skoog (MS) medium for shoot regeneration and rooting with 6−BA (BAP—6−benzylaminopurine) and ABA (abscisic acid) as the major growth regulators and kanamycin (50–150 mg L^−1^) as the selective agent. However, many other factors varied widely in proposed ‘optimized’ protocols, which probably reflects the genotypic effects or experimental conditions among those experiments. The reported overall transformation efficiency varied among different cucumber genotypes. The US slicing cucumber Poinsett 76 (PS76) and several North China−type (Chinese Long) cucumber lines such as Changchunmici (CCMC) or Xintaimici (XTMC) seem to be less recalcitrant to *Agrobacterium*−mediated transformation and are popular choices in various transformation studies (reviewed in [15]). While cucumber plant regeneration can be achieved from somatic embryogenesis via callus culture [17,18,19,20]), direct organogenesis seems more efficient and time saving ([17,21]). For direct organogenesis, most cucumber transformation studies used cotyledon nodes or hypocotyls as the sources of explants (e.g., [22,23,24,25]). Early identification of transformants has been facilitated with reporter genes allowing GUS staining (e.g., [19,23,26]) or GFP fluorescence detection (e.g., [27,28,29]). Inspired by work from other crop plants, additional measures have been explored aiming to enhance *Agrobacterium* infection including vacuum infiltration, sonication, micro−brushing of explants, or addition of antioxidant chemicals in culture media to reduce oxidative stresses caused by mechanical wounding (e.g., [29,30,31]). 

Despite the significant progresses made in *Agrobacterium*−mediated cucumber transformation in recent years, the transformation efficiency (TE) reported in different studies was generally low but varied widely between 0.1% and 5% (see the Discussion section below for more details). It seems that *Agrobacterium*−mediated transformation in cucumber, to some extent, remains an art. This could be seen in the very different TEs even from the same genotype. In many cases, the reported transformation protocols lack necessary experimental details for others to repeat. Some protocols used specific cucumber genotypes or plasmid vectors that are not readily accessible by the community. A reliable and reproducible protocol with predictable transformation efficiency evaluated with consistent or comparable criteria is lacking. Thus, one motivation of the present study was to evaluate some important factors affecting cucumber transformation efficiency and develop a baseline protocol that is amendable for further optimization by the community. Here we reported the effects of cucumber genotypes, explant type and age, *Agrobacterium* strains, and antibiotics on shoot regeneration and the overall transformation efficiency, which were exemplified using two target genes (*LL* and *APRR2*) in the US slicing cucumber line Poinsett 76 (PS76) and the US pickling cucumber line H19. The protocol was also tested by developing transgenic plants in the 9930 background for three additional genes including the broad−spectrum disease resistance gene *CsSGR* (*staygreen*) [32], the fruit shape gene *FS1.1,* and the target leaf spot (TLS, causal agent *Corynespora cassiicola*) resistance gene *Cca4*.

One important use of *Agrobacterium*−mediated transformation in plants is the generation of T−DNA insertional mutants [33,34,35]. In Arabidopsis and rice, tens of thousands of T−DNA insertion lines have been developed, which has provided a powerful tool to investigate gene functions through reverse genetics (e.g., [36,37,38]). T−DNA is integrated into the host plant genome randomly via illegitimate recombination, which may result in mutagenesis in non−target genes [39]). Theoretically, in any *Agrobacterium*−mediated transgenic plants, mutant phenotypes should be observed if the T−DNA insertion is inside a gene responsible for a phenotype. Reports on T−DNA insertion mutants in cucumber transgenic plants are rare (e.g., [40]). Thus, the second objective of the present study was to assess the potential of T−DNA insertion mutagenesis as a tool to generate mutants, which was exemplified with four confirmed mutants for leaf color and fruit size/shape. 

## 2. Materials and Methods

### 2.1. Plant Materials and Growth Conditions

Five cucumber inbred lines from different market groups were used for assessment of effect of genotypes on transformation efficiency. They included one US slicing cucumber—Poinsett 76 (PS76), two US pickling type lines—Gy14M and H19, and two north China fresh market (Chinese Long, CL) lines—9930 and WI7602. Seeds of 9930 and WI7602 were kindly supplied by Xingfang Gu (Chinese Academy of Agricultural Sciences, Beijing, China) and Sen Li (Shanxi Agricultural University, China), respectively. Gy14M (also known as Gy14M42) is the monoecious version of the gynoecious Gy14 [41]. H19 is a *littleleaf* (*LL*) mutant [25]. Genome assemblies for both Gy14 and 9930 are publicly available at http://cucurbitgenomics.org/v2, (accessed on 26 February 2023). All lines are monoecious with typical fruit morphological features of their market groups [1]. Plantlets regenerated from tissue culture were kept in jars with MS medium in a tissue culture chamber with a temperature cycle of 28 °C/18 °C and 16 h/8 h of light/dark. At approximately the 5−true−leaf stage, the seedlings were transplanted into pots in a greenhouse. 

### 2.2. Plasmid Vectors and Agrobacterium Strains

We used two cucumber genes as examples to evaluate the transformation efficiency and mutagenesis; these included *littleleaf* (*LL*), which encodes a WD40 repeat domain−containing protein regulating organ size variation [25]; and *APRR2* (*Arabidopsis pseudo−response regulator 2*), which underlies the *w* locus for white immature fruit color [42,43]. Transgenic plants overexpressing (OE) the wild−type allele of the *LL* gene that was driven by the CaMV 35S promoter and RNAi (RNA interference) knockdown line for the *APRR2* gene were developed. For OE, the coding sequence (CDS) of the *LL* gene was cloned and inserted into the plasmid of pCAMBIA2301−ky digested with *Kpn* II and *Xba* I to generate the *Pro35S:LL* construct (LL−OE construct in short hereinafter) [25,44]. The RNAi construct for *APRR2* was generated using two gene−specific fragments that were inversely inserted into the pFGC1008 plasmid vector double digested with *Asc* I/*Swa* I and *BamH* I/*Spe* I (RNAi−APRR2 construct in short hereinafter) (Appendix A) [45,46]. Identity of all constructs was verified by Sanger sequencing; the construct was then introduced into *Agrobacterium* via electroporation for the AGL1 strain and via the freeze–thaw method for GV3101 [47]. Transformation of OE vectors was performed in both PS76 and H19 for comparison of the transformation efficiency between genotypes, while RNAi was conducted in PS76 with AGL1 and GV3101 to assess effects of *Agrobacterium* strains on transformation. 

### 2.3. Cucumber Transformation Procedures

The entire transformation process was divided into 14 main steps, which are illustrated in Appendix A. Step−by−step protocols of the entire procedure (including the formulae and components of all growth media used in this study) are provided in Appendix A. Briefly, seeds were soaked in tap water for 60 min and then peeled, surface sterilized by soaking in 75% ethanol for 1 min, and rinsed 3 times with autoclaved ddH_2_O. Then, the seeds were sterilized one more time with 6% sodium hypochlorite (NaClO) for 12 min followed by rinsing with ddH_2_O three times. Seeds were dried by blotting on sterilized filter paper and then placed on MS germination medium for 3 days. 

Explants were dissected from cotyledon nodes and/or hypocotyls of germinating seeds with forceps and a tweezer then placed in sterilized ½ MS liquid medium or water to avoid drying out before inoculation. The *Agrobacterium* suspension was prepared by shaking a single colony in Luria–Bertani (LB) medium for about 48 h to reach OD_600_ = 0.7. Then, acetosyringone (AS, 200 μM) and Silwet L−77 (0.05%) were added to the bacterial suspension. The explants were immediately transferred to the *Agrobacterium* solution with gentle shaking by hand for 12 min. In some trials, we also tested vacuum infiltration to enhance bacterium infection (Appendix A) [29,30]. Nevertheless, we did not find a significant improvement in the transformation efficiency from vacuum infiltration, which may also have increased the chance of contamination. After pouring off the inoculation liquid, the explants were blotted dry with sterilized filter paper. Then, the explants were placed onto a piece of sterilized filter paper in a Petri dish containing co−culture MS medium. The explants were co−cultured with *Agrobacterium* in the dark at 23 °C or 28 °C for 2 d and transferred to the differentiation MS medium with appropriate antibiotics and hormones. The explants with a yellowish color were subcultured every 2 weeks until the shoots had 4–5 small green leaflets. The shoots with green leaves were then transferred into the rooting medium to induce roots. When 3 or more radicles or main roots were visible, the plantlets were transplanted to the soil in small plastic cubes covered with a plastic dome or Saran Wrap film to maintain humidity and acclimation. Hardy T_0_ plants were transplanted into large pots and grown in greenhouses. Self−pollinations were conducted on these plants to obtain T_1_ seeds.

### 2.4. Treatments with Phytohormones and Antibiotics

No hormones were added to the germinating media. To assess the effects of synthetic cytokinin and ABA on shoot regeneration, four concentrations of 6−BA (2.0, 1.5, 1.0, and 0.5 mg L^−1^) and two concentrations of ABA (1.0 and 0.5 mg L^−1^) (a total of eight combinations) were tested with five cucumber lines. The antibiotic selective efficiency was evaluated with two antibiotics: kanamycin with three treatments (0, 50, 100 mg L^−1^) and hygromycin with four treatments (0, 5, 10 and 15 mg L^−1^) in the shoot−regeneration growth medium. For each antibiotic, the concentration in the selective rooting medium was decreased to half of that in the shoot−regeneration medium. The shoot−regeneration rate was defined as the number of explants with shoot regeneration divided by total number of explants used multiplied by 100. 

### 2.5. PCR and qPCR Verification of Transformants and GUS Activity Assay

To verify antibiotic−selected putative transgenic plants, leaf samples were isolated from emerging new leaves of plants grown in the greenhouses to avoid possible contamination by residual *Agrobacterium*. Genomic DNA was extracted with the CTAB method. PCR was performed by cloning a 194 bp fragment of the *npt II* gene (for kanamycin resistance) from the T−DNA region in the plasmid vector (primer sequences: forward 5′CTCTGATGCCGCCGTGTTCC 3′; reverse 5′CGCCCAATAGCAGCCAGTCC 3′). Quantitative real−time PCR (qPCR) was used to examine the expression level of the *LL* gene in LL−OE transgenic plants. Total RNA was isolated with the plant RNA purification mini kit, and cDNA was synthesized with a reverse−transcription kit (both from Thermo Fisher Scientific, Waltham, MA). The qPCR was performed on a QuantStudio™ 3 Real−time PCR system following our early procedure ([25]) and using the cucumber *CsActin* gene as the internal reference ([32]). The relative expression level was calculated using the 2^−ΔΔCT^ method. For qPCR of each sample, there were three biological and three technical replicates. Significance was determined via pairwise *t*−tests. The overall transformation efficiency (TE, %) was defined as the numbers of PCR−positive plantlets from independent transformation events divided by total number of explants or seeds used multiplied by 100. 

The *GUS* activity (for β−glucuronidase) in the seeds of the putative transformant was measured with MUG (4−methyl−umbelliferyl−β−d−glucuronide, Sigma−Aldrich) staining according to a previously described method [48].

### 2.6. Identification and Characterization of Morphological Mutants

LL−OE transgenic T_1_ plants together with their wild type (WT) PS76 were grown in the University of Wisconsin Walnut Street Greenhouses facility (WSGH) or in the field of the Hancock Agricultural Research Station (HARS). Throughout the growing season, all plants were carefully examined for any variation from the WT. These traits included the color, shape, and size of the roots, leaves, stem, flowers, and fruits. Measurements of the size of fruits, leaves, stems, and flowers were conducted with rulers in replicated trials. Plants showing any non−target gene−related morphological variation were tagged and self−pollinated to advance to T_2_, which were planted again to verify the inheritability of the mutant traits. Putative mutants were crossed with other WT cucumber lines to develop segregation populations to investigate the inheritance mode of the mutant. Some mutants were observed in both greenhouses and open fields to assess the effect of environments on expression of the mutant phenotype.

## 3. Results

### 3.1. Evaluation of Cotyledon and Hypocotyl as Explants for Direct Shoot Organogenesis

Shoots can be regenerated directly from explants or indirectly from calli. While each method has its own advantages and disadvantages, direct regeneration of shoots from the cotyledon node is the most popular method in cucumber transformation ([17,21,30,46,49]). In a germinating seed (Figure 1A), meristematic tissues for culture could be from the shoot apical meristem (SAM) or subapical ground meristematic region in the cotyledon–hypocotyl junction [50] (Figure 1B). The SAM region is small and grows quickly. It is difficult to mechanically wound and lacks a key surface component for recognition by *Agrobacterium* (Figure 1C,D) ([51,52]). Indeed, we collected the SAM region from the germinating seeds of PS76 and inoculated them directly with *Agrobacterium* carrying the LL−OE construct. Of 246 explants inoculated and co−cultivated, all were able to generate shoots quickly in regeneration medium, but no resulting plantlets were able to survive from the selection with 100 mg L^−1^ kanamycin (Appendix A). On the other hand, the ground meristem grows slower than SAM, and it is relatively easy to generate mechanical wounding during dissection of the explants (Figure 1D–I). Thus, both cotyledons and hypocotyl containing ground meristem cells were tested as explants. In addition, each cotyledon piece could be cut transversely into two parts: the proximal half and the distal half (Figure 1F–G); the proximal half could be cut twice to make a V−shaped wound [23](Figure 1H) or cut longitudinally into two pieces (Figure 1I) [30]. In this way, there could be a maximum of four cotyledon explants from one seed. 

We also evaluated the use of hypocotyls as explants [24,53]. We excised the proximal part of the hypocotyl of PS76 that carried the LL−OE construct (Figure 1D) for *Agrobacterium* inoculation/co−cultivation. As compared with cotyledon explants, hypocotyl explants took less time for shoot regeneration. One challenge in using hypocotyl as the explant was to isolate the proximal hypocotyl part under a dissecting microscope to avoid any residual SAM, which was time−consuming and prone to contamination. To address this potential issue, we cut the hypocotyl at the position a bit far away from the SAM without using a microscope (Figure 1E). From 719 hypocotyl explants (719 seeds) tested, we obtained three PCR−positive LL−OE transformants (TE = 0.42%, Appendix A), which was similar to the 0.5% TE when cotyledons were used as the explants, in which 21 PCR−positive LL−OE transformants were obtained from 4206 explants (from 2103 seeds). This suggested that both the cotyledon and distal hypocotyls could be good sources of explants. 

### 3.2. Effects of Explant Age and Co−Cultivation Temperature/Duration on Shoot Regeneration 

After co−cultivation with *Agrobacterium*, we found that cotyledon explants displayed either a yellowish or whitish color (Figure 2). When cultured in shoot−regeneration medium, shoots could only be generated from yellowish explants, while none could be produced from whitish ones. The frequency of whitish−colored explants seemed to be correlated with the age of explants (days of germination of seeds). When the seeds of PS76 were geminated at 28 °C for 2 d (~2 cm total length) and then the excised explants were co−cultured at the same temperature for 2 d, ~40.5% of the explants (17/42) showed a whitish color (Figure 2A,B). When the cotyledon explants from seedlings after 3 d of germination at 28 °C in the dark (~4 cm in length) were co−cultivated with *Agrobacterium* for 2 d, approximately 13.0% (7/54) of the explants were whitish (Figure 2C,D). Bacterium co−cultivation also increased the frequency of whitish cotyledon explants to 1.4% (1/70) (Figure 2E) and 14.6% (Figure 2F) when co−cultured in the dark for 2 d without and with inoculation with *Agrobacterium*. 

Previous studies indicated that lower temperatures (18–20 °C) during co−cultivation may increase the transformation efficiency (e.g., [54]). We tested the effect of two temperature settings (23 °C and 28 °C) of co−cultivation on shoot−regeneration efficiency. Of 160 cotyledon explants from PS76 planted in shoot−regeneration medium with 50 mg L^−1^ kanamycin, 36 (22.5%) and 22 (13.8%) yielded shoots at 23 °C and 28 °C, respectively (Appendix A), which supported our earlier observation that a lower temperature promoted shoot regeneration.

### 3.3. Effect of Agrobacterium Stains, Antibiotics, and Selection Pressure on Shoot Regeneration or Transformation Efficiency

Several *Agrobacterium* strains such as AGL1, GV3101, EHA105, EHA101, and LBA4404 have been used in cucumber transformation [15], but there is no report that compared the transformation efficiency of different strains. We compared the transformation efficiency between AGL1 and GV3101. We developed RNAi−APRR2 transgenic plants in PS76 using both strains (Figure 3). A total of 1000 explants (from 500 seeds) were co−cultured with each strain (Figure 3). One and three independent PCR−positive transgenic plants were obtained with GV3101 (TE = 0.10% with explants or 0.20% with seeds) and AGL1 (TE = 0.30% with explants or 0.60% with seeds), respectively, suggesting a significantly higher efficiency of AGL1−mediated transformation than GV3101 in PS76.

During inoculation/co−cultivation with *Agrobacterium*, only cells close to the wounding sites could be successfully infected [28]. Non−transformed or chimeric shoots or plantlets could be produced during tissue culture. Antibiotics are often added to the growth medium as a selective agent to kill or inhibit growth of untransformed cells or tissues. The T−DNA in many plasmid constructs carry the *neomycin phosphotransferase II (npt II*) gene conferring kanamycin resistance [55]. In PS76 cucumber, 50–100 mg L^−1^ kanamycin was widely used to select positive transformants [21,23,27]. In developing LL−OE transgenic plants, we evaluated the survival rate of PS76 explants in culture media containing 0, 50, and 100 mg L^−1^ kanamycin. In kanamycin−free culture medium, 41.3% of the explants (*n* = 75) could produce shoots, and the rate decreased to 16.9% (*n* = 71) and 4.2% (*n* = 310) when the shoot−regeneration medium was supplied with 50 and 100 mgL^−1^ of kanamycin, respectively (Figure 4A–C). At 100 mg L^−1^ of kanamycin, most shoots that initiated from the explants had leaves that were yellow or turned to yellow at later stages and stopped growing; only plantlets with green leaves could continue growth during subcultures (Figure 4D,F). In chimeric plants, non−transformed leaves could also be inhibited or killed by kanamycin during subculturing (Figure 4G). 

Roots were much more sensitive to kanamycin toxicity than shoots: at 100 mg L^−1^ of rooting medium, only very few roots could be initiated from the plantlet; at 50 mgL^−1^, the shoot could produce 4–5 main roots, which was still fewer than the root numbers in kanamycin−free medium (Figure 4H). At 50 mg L^−1^ of kanamycin, false−positive transformants could be effectively eliminated (Figure 4I). The use of kanamycin in both shoot and root regenerating media saved time and labor for subsequent PCR validation and transgenic plant growth.

During development of RNAi−APRR2 transgenic plants in PS76, the pFGC1008 vector, which contains the selective marker *hyg* for hygromycin resistance, was used (Appendix A). Thus, hygromycin was added to the culture medium as a selective agent. We found that PS76 was very sensitive to hygromycin. Almost no shoots could be initiated from explants from the culture medium with >10 mg L^−1^ hygromycin; all explants were strongly chlorotic or albino at 15 mg L^−1^. Shoots could be initiated at 5 mg L^−1^ of hygromycin (Appendix A). At this concentration, from 1000 explants, we were able to generate four plantlets, three of which were validated by PCR with an estimated transformation efficiency of 0.30% (=3/1000 × 100)**.**

### 3.4. Genotypic Effects on Shoot−Regeneration Rate and Transformation Efficiency

We compared the shoot−regeneration efficiency from cotyledon explants in Gy14M, 9930, H19, PS76, and WI7602 (Figure 5). Previous studies have shown that 6−BA and ABA are the most important phytohormones affecting shoot regeneration in cucumber transformation [22,23,31,56,57]. To evaluate genotypic effects, we first optimized the concentrations of the two hormones. Eight treatments of the shoot culture medium were set up from combinations of four concentrations of 6−BA (at 0.5, 1.0, 1.5, and 2.0 mg L^−1^) and two concentrations of ABA (at 0.5 and 1.0 mg L^−1^). For each treatment, >40 cotyledon explants from each genotype were tested. Representative images for each line with the highest regeneration rate at a given treatment are shown in Appendix A. The effect of 6−BA/ABA treatments on shoot regeneration varied among different cucumber lines and was more obvious in the American picking (Gy14M and H19) and slicing cucumber (PS76) lines. The average shoot−regeneration rate across all eight treatments for 9930, WI7602, PS76, H19 and Gy14M was 87.9, 68.1, 29.6, 19.8, and 17.5%, respectively (Appendix A). The corresponding highest regeneration rates were 97.8, 77.8, 52.1, 42.2, and 33.3%, which were observed at different 6−BA/ABA combinations among these lines (Figure 5). Overall, the shoot−regeneration ability was significantly higher in 9930 and WI7602 than in PS76, H19, or Gy14M (Appendix A). These data suggested that Chinese Long cucumbers have a higher shoot−regeneration rate than the three US picking or slicing cumbers. Nevertheless, under the optimal hormone concentrations, PS76 exhibited a comparable shoot−regeneration efficiency to WI7602, which was consistent with earlier work. 

We further tested the protocol by developing transgenic lines in the 9930 genetic background for three target genes including overexpressing the candidate gene for the fruit shape QTL *FS1.1* and the TLS resistance gene *Cca4* (unpublished data), as well as CRISPR−Cas9−based gene editing of *CsSGR* (*dm1/psl/cla1*), a gene that contributes to broad−spectrum resistances against three pathogens [32]. For *FS1.1* and *Cca4*, we employed the same set of parameters that we used for LL−OE in PS76. To develop transgenic plants overexpressing *FS1.1*, we harvested two cotyledon explants from each germinating seed of 9930. Out of 1,200 explants from 600 seeds, we identified 3 verified, independent transgenic lines with a TE of 0.25% (based on the number of explants; 0.50% based on the number of seeds). For *Cca4*, we obtained 5 PCR−positive plants from 1,200 cotyledon plants (300 seeds, 4 explants per seed) with an estimated TE of 0.42% (based on the number of explants harvested) or 1.67% (based on 300 seeds used). The lower TE (0.25%) for *FS1.1* than that for *Cca4* (0.42%) based on the number of explants was interesting. It is not known if more explants harvested from a germinating seed (four vs. two in this case) will affect the shoot−regeneration ability, and this may require further investigation. In developing CRISPR−Cas9−edited *CsSGR* mediated by EHA105, from 4100 cotyledon explants (from ~2000 seeds, 2 explants per seed), we obtained 23 PCR−positive plants with an estimated TE of 0.56% (based on 4100 of explants harvested) or 1.12% (based on 2050 seeds used). We averaged the TE by genotype from various experiments from this study for PS76, H19, and 9930; the results were 0.90, 0.50, and 1.05% (based on the number of seeds) or 0.40, 0.25, and 0.48% (based on the number of explants), respectively. The TE for individual experiments is presented in Appendix A. The overall transformation efficiency seemed largely consistent with the shoot−regeneration rate (Figure 5) among the three lines. 

### 3.5. Additional Evidence for Validation of Positive Transformants

During shoot regeneration, subculturing, and rooting, the addition of selective antibiotics in the culture medium allowed for the elimination of most false positives (Figure 4). True transformants could be further validated with PCR amplification of a fragment from the T−DNA in a vector (e.g., *npt II*) from newly emerged leaves of putative transgenic plants (e.g., see Figure 3C,F and Figure 6C). However, the most direct evidence should be from the performance of the target genes in transgenic plants. Various approaches could be taken to functionally characterize the transgenic plants depending on the nature of the transgenic events. For the LL−OE transgenic plants, we examined the expression level of *LL* in 10 independent T_1_ OE plants (OE1−OE10) with qPCR and found that OE6 and OE7 exhibited a significantly higher expression of *LL* that was more than a 30−fold increase compared with the control; these were followed by OE5, OE9, and OE10, which were 3–5 times more than the control (Figure 7A). For OE5, OE7, and OE10, we performed GUS assays in the young developing seeds (T_1_) that exhibited the characteristic blue color, suggesting a successful integration of the T−DNA into the PS76 genome in these OE plants (Figure 7B). When the OE (OE1, 5, 7 and 9; T_1_) and WT seeds were germinated in MS medium containing 100 mg L^−1^ kanamycin, all OE seedlings were able to develop longer and more numerous roots than the WT (Figure 7C). The *LL* gene regulates organ size in cucumber [25]. The increased leaf size and other organs in these T_1_ OE lines provided the most direct evidence of successful transformation of the *LL* gene (Figure 7D). 

For transgenic plants, inheritability of the target gene is important for gene function studies or crop improvement. The poor fertility in early−generation transgenic plants has been observed in many plant species (e.g., [13,58,59,60,61]). This seemed particularly true for the LL−OE plants in both the PS76 and H19 backgrounds in the present study. In most independent T_0_ lines, we found very few fruits with plump seeds inside. For example, in PS76−LL−OE, only 2 and 12 plump seeds were harvested from the fruits of OE6 and OE12, respectively (Appendix A). Fruits from OE7 and OE8 T_0_ plants had no developed seeds, and no plump seeds could be obtained from OE−H19−LL−1 T_0_ plants. Eventually, we were able to obtain several seeds from OE7 and OE8 (but not OE−H19−LL−1) by growing a cutting branch from the original T_0_ plants. The poor fertility in most T_0_ transformants often could be restored at least partially in subsequent generations [62]. However, among four independent PS76−RNAi−APRR2 T_0_ transgenic plants, three were sterile with no seeds, and one produced only three plump seeds (Appendix A). After five more generations of self−pollinations, the resulting T_5_ RNAi−APRR2 plants still had very low fertility, which was also true for the F_1_ plants from reciprocal crosses between the T_5_ RNAi plant and PS76 (Appendix A).

### 3.6. Identification and Characterization of Putative Mutants from T−DNA Insertions 

During transformation, the locations where T−DNA integrates into the recipient genome are largely random [63], which can result in T−DNA insertion mutation. In the phenotypic characterization of the transgenic OE−LL plants, we identified a dozen mutants with morphological variation as compared with the WT (PS76). The phenotypes of four such mutants are exemplified in Figure 8 and include *yellow flesh and skin* in mature fruits, which was in contrast with the light−green mature fruit and white flesh in PS76 (Figure 8A,B), *short and slim fruit* (Figure 8C,D), *short and fat fruit* (Figure 8E,F), and *yellow leaf* (Figure 8G–I). The *yellow leaf* mutant showed a similar green leaf color to PS76 in the greenhouse, but in the field, the green true leaves gradually turned to yellow, suggesting a possible development stage and/or environment dependent regulation of leaf color in this mutant (Figure 8I). All seemed inheritable because they showed consistent mutant phenotypes in both T_1_ and T_2_ generations. Further, we developed an F_2_ segregation population from the cross between Gy14M and the *yellow leaf* mutant. Among 581 F_2_ plants examined, 439 and 142 had green and yellow leaves, respectively, which fit a 3 green:1 yellow segregation ratio (χ^2^ = 0.09 and *P* = 0.76), which suggested that a single recessive nuclear gene controls the *yellow leaf* mutant phenotype. 

In addition to the above four mutants, we also isolated a number of other morphological mutants from the progeny of PS76−OE−LL T_0_ plants. A few examples are shown in Appendix A; these include *wrinkled leaf* (Appendix A) and *jagged edge* but flat leaf (Appendix A). Interestingly, we intended to overexpress the *LL* gene to enlarge the leaf size, but one plant was a dwarf and had very *small leaf* size (Appendix A). The wild type PS76 is monoecious, but we found a *gynoecious* plant (Appendix A). The inheritability of these mutant phenotypes requires further investigation.

## 4. Discussion

### 4.1. Agrobacterium−Mediated Transformation Efficiency in Cucumber

Since the first report of cucumber genetic transformation [13], numerous studies have been conducted that aimed to improve the transformation efficiency (TE) (reviewed in [14,15,16]). A consensus from early investigations is that cucumber, like other cucurbit crops, is recalcitrant to *Agrobacterium*−mediated transformation and has a much lower TE than many other crops outside the Cucurbitaceae family. Further, the TE reported in different studies varied significantly. In the literature, two methods were used to estimate TE. In some studies, the TE was defined as the number of T_0_ plants out of the total number of start explants. These putative T_0_ transgenic plants were verified by multiple approaches that included Southern hybridization, PCR, and phenotypic characterization of target genes in T_1_ or T_2_ plants. Using these criteria, [23]) reported a TE of ~1.7% in PS76, which was 1.1% in the cucumber genotype Greenlong [26]. In the South China−type cucumber line Cu2, three independent T_0_ transgenic plants were generated from 1132 seeds with an estimated TE of 0.27% [28] (if one explant from one seed is assumed). In the present study, using the number of explants as the base of calculation in the RNAi−APRR2 transformation experiments with PS76, we estimated a TE of 0.10% and 0.30% for the *Agrobacterium* strains GV3101 and AGL1, respectively. The TE for LL−OE was 0.25% in H19 and 0.49% in PS76. In the 9930 background, the transformation efficiency for *FS1.1*, *Cca4*, and *CsSGR* was 0.25, 0.43, and 0.56% respectively. The average TE by genotype for PS76, H19, and 9930 from different experiments of this study was 0.90, 0.50, and 1.05% (based on the number of seeds) or 0.40, 0.25, and 0.48% (based on the number of explants), respectively (see Appendix A). 

In a few studies, GFP was used as the reporter gene and the TE was calculated as the number of GFP−positive shoots [27,30] or T_0_ plants [29] out of the total number of explants. In this way, Nanasato et al. [30] found that the transformation efficiency varied from 7.5–16% with an average of 11.9 ± 3.5% in the Japanese cucumber line Shinhokusei No. 1. In PS76, the TE was 21% (*Agrobacterium* strain EHA105) or 8.5% (strain LBA4404) [27]. Via the combined use of optimized vacuum infiltration, micro−brushing, sonication, and the addition of antioxidants, Xin et al. [29] reported a TE of 5.18, 2.20, 1.97, and 2.46% for Cu2, XTMC, 404 (Chinese Long), and Eu1 (European−type) cucumbers, respectively. The TE based on GFP−positive explants was probably an overestimate of the actual TE (see discussion below). However, due to the different methods used, the TEs reported in different studies were not readily comparable. Other than the many factors discussed in the study, the experiences of the researchers probably also contributed to the varying TE reported. From a practical perspective, TE estimation based on the number verified/validated T_0_ or advanced−generation transgenic plants is more useful and should be used to estimate the TE in future studies. Since multiple explants could be obtained from a single seed, the number of seeds and explants used in a study should also be reported for easy comparison. 

### 4.2. Effects of Cucumber Genotypes on Transformation Efficiency

The varying transformation efficiency among different studies discussed above could also be attributed to the different cucumber lines used. The shoot−regeneration rate and overall TE in cucumber are known to be genotype−dependent (e.g., [15,16,64,65,66]). In the present study, we compared the shoot−regeneration efficiency among PS76 (US slicer), Gy14M (US pickle), H19 (US pickle), 9930 (Chinese Long), and WI7602 (Chinese Long). Across eight 6−BA/ABA treatments, the average shoot−regeneration rate for 9930 and WI7401 was significantly higher than that of the other three lines; 9930 (87.9%) and Gy14M (17.5%) had the highest and lowest mean regeneration rate frequency, respectively (Figure 5). These data were largely consistent with the notion that Chinese Long cucumbers have an overall higher regeneration rate than cucumbers in other market groups. In addition, based on the data of multiple experiments in this study, the mean transformation efficiency of 9930 was also higher than that of PS76 and H19 (Appendix A), indicating a positive correlation between the shoot−regeneration rate and the overall transformation efficiency. However, the transformation efficiency can vary even within the same market group (e.g., [29,31]). In reality, the ability to select a specific line for *Agrobacterium*−mediated transformation may be limited and in many cases may be dictated by the traits under investigation or the research objectives. Nevertheless, both PS76 and 9930 should be the genotypes of choice for *Agrobacterium*−mediated transformation studies. 

One interesting question regards the genetic or molecular basis of different transformation efficiencies among cucumber genotypes. In an earlier study, QTL mapping was conducted to identify the QTL for shoot−regeneration ability using a recombinant inbred line (RIL) population derived from 9930 (high efficiency) and 9110Gt (low efficiency) [57,67]. Four QTLs were identified that explained 9.7–16.6% of the observed phenotypic variance. Further, a candidate gene (*Csa1G642540*) was proposed for one major−effect QTL that was a homolog of Arabidopsis *AT3G44110* encoding DnaJ Homologue 3 (J3). While the linking of this gene to shoot−regeneration ability requires further evidence, this work did show that there is a genetic basis for the ability of organ/tissue/plant regeneration in cucumber.

In Arabidopsis, many development regulator genes have been identified that play critical roles in tissue/organ/plant regeneration, callus formation, or wounding response and repairing. Some well−characterized genes are *WUSCHEL* (*WUS*), *PLETHORAS (PLT)*, *BABY−BOOM (BBM)*, *ENHANCED SHOOT REGENERATION (ESR)*, *GROWTH REGULATING FACTORS (GRF)*, *GRF−INTERACTING FACTOR* (*GIF*), and *WOUND INDUCED DEDIFFERENTIATION 1* (*WIND1*). Overexpression of these genes is often associated with improved callus formation and/or plant regeneration, while deficient mutants of these regulators significantly decrease callus formation and shoot regeneration [68,69,70]. Introduction of constructs carrying *PLT5*, *WIND1, WUS,* or their combinations significantly increases the in planta transformation efficiency in multiple crops including snapdragons, tomato, and *Brassica rapa* [71]. The expression of a fusion protein combining wheat *GRF4* and its cofactor *GIF1* substantially increased the efficiency and speed of regeneration in wheat, triticale, and rice and increased the number of transformable wheat genotypes [72]. Overexpressing a chimeric fusion of *ClGRF4* and *ClGIF1* increased the transformation efficiency in watermelon [73]. 

We wondered if any of these development regulator genes may be associated with the varying transformation efficiencies in different cucumber lines. From the Gy14v2.1 and 9930v3.0 genome assemblies (http://cucurbitgenomics.org/v2, accessed on 26 February 2023), we identified the closet homologs of the eight Arabidopsis development regulator genes: *CsESR1* (*CsGy2G017320*)*, CsWIND1* (*CsGy2G016460*)*, CsWUS* (*CsGy6G031220*)*, CsPLT5* (*CsGy1G020130*), *CsBBM* (*CsGy2G008180*), *CsGRF4* (*CsGy2G026820*), *CsGIF1* (*CsGy2G002810*), and *CsGRF5* (*CsGy3G026820*). We extracted the complete genomic DNA sequences (including the promoter region of each gene) from the PacBio genome assembly of Cuc2 [29,74] and HIFI genome assembly of PS76 (unpublished data). Among 9930, Gy14M, Cuc2, and PS76, the 9930 and Gy14M varieties had the highest and lowest shoot−regeneration frequency, respectively, and PS76 was in the middle (Figure 5). For each gene, we aligned the gDNA sequences of the four lines, but no consistent polymorphisms or haplotypes were found among them that could be potentially associated with the shoot−regeneration efficiency. Nevertheless, this merits more investigations in the future to find a possible correlation of the expression of these genes with the transformation efficiency. It is possible that overexpression of any or some of these genes may boost the success rate of shoot regeneration or the overall transformation efficiency.

### 4.3. Effect of Explant Types and Ages on Transformation Efficiency 

In cucumber transformation, most investigations used cotyledons as the explant source and collected one or two explants per germinating seed. Nanasato et al. [30] tested the method to cut one cotyledon into two pieces, which allowed four explants from one seed. In this study, we found the proximal part of hypocotyl could also be a source of explants (Figure 1). We compared the transformation efficiency using explants from both cotyledons and the hypocotyl and found that both sources of explants had a similar efficiency. Thus, up to five explants could be harvested from one germinating seed (Figure 1). This may be an advantage when seed availability is an issue. This method may also save time in explant preparation. 

The age of cotyledon explants taken from the germinating seeds seemed to affect shoot regeneration significantly (Figure 2). After two days of co−cultivation with *Agrobacterium* in the dark, the explants could be either yellowish or whitish. The whitish explants were not able to regenerate into shoots (Figure 2). Explants from seeds germinated at 28 °C for 2 d had more (~41%) with a whitish color than those from seeds germinated at the same temperature for 3 d (13%). In addition, co−cultivation also resulted in more whitish explants (14.6%) than those without (1.4%) inoculation with *Agrobacterium* (Figure 2E,F). If the explants were harvested from seedlings after 5 d of germinating seeds, the cotyledons were too thin to generate shoots. This suggested that for PS76 and 9930, germinating the seeds for 3 d might be appropriate for harvesting the explants. However, the seed germination rate may be influenced by multiple factors such as age and maturity, and the optimal time for a specific genotype could be identified from a preliminary study based on the frequency of yellow explants after co−cultivation. 

In this study, we found that the frequency of yellow explants after co−cultivation was positively correlated with the shoot−regeneration rate (Figure 2). This was similar to what was observed in the genetic transformation of *Brachypodium distachyon,* in which calli could be either white or yellow and only yellow calli had the ability to transform [75]. The healthy yellowish color was probably the result of carotenoid accumulation. High levels of carotenoids can be produced in the etioplasts of seedlings that germinate in the dark, which gives the cotyledons their characteristic yellow color [76]. In dark−grown Arabidopsis, enhanced production of carotenoids in plastids improves transition to photosynthetic development upon exposure to light (photomorphogenesis). In the green leaves, carotenoids play important roles for photosynthesis and photoprotection [77,78]. If the explants are harvested too early or too late from the germinating seeds or are inoculated with *Agrobacterium*, they may be less tolerant to the stresses due to wounding or pathogen infection, which may result in a whitish color due to less carotenoid accumulation (and thus less protection). Of course, the white explants could also be due to the use of poor or underdeveloped seeds.

### 4.4. Increase Agrobacterium Infection and Efficiency of Selection for Positive Transformants

For successful *Agrobacterium*−mediated transformation, effective bacterial infection of the explants and selection of transformed cells is critical. Many factors play roles in these processes including bacterial strains, infection or co−cultivation time/duration, and selective antibiotics and selection pressure [15,16]. *Agrobacterium* strains have been shown to be a main factor that influences the transformation efficiency. In *Medicago truncatula*, the transformation efficiency with the hypervirulent strain AGL1 was twice that obtained with LBA4404 [79]. AGL1 also exhibited a higher efficiency than C58, GV3101, and EHA105 in switchgrass [80], but it was lower than GV3101 in tomato [81]. In cucumber, several *Agrobacterium* strains such as C58, GV3101, and EHA105 have been used in genetic transformation, of which EHA105 seems to be the most often used (e.g., [46,82,83,84]). In this study, we compared the transformation efficiency between two strains: AGL1 and GV3101. Consistent with work in other plants, we found that AGL1 (TE = 0.30%) had a higher transformation efficiency than GV3101 (TE = 0.10%) (Figure 3). We did not observe any obvious negative effects on explants or shoot regeneration associated with the more virulent AGL1. As such, AGL1 seems to be a good choice in *Agrobacterium* cucumber transformation studies. 

Due to the large amount work in transformation, an efficient selection system is critical to eliminate untransformed cells and reduce false positives and chimeras, thus reducing the workload. Some studies employed reporter genes such as GUS or GFP to detect infected/transformed cells (e.g., [19,23,26,27,28,40]). The uses of a GFP reporter system or antibiotic selection for true transformants have their own pros and cons. The GFP reporter system is amendable for high−throughput screening of positive transformants. When the GFP reporter gene is used for positive shoot selection, several shoots could be initiated from the explant. If no selection pressure is applied, GFP−positive shoots might be chimeric with a mixture of transformed and untransformed cells (e.g., [27,30]). It is time−consuming to identify them at T_1_ generation. In addition, it is not known how many true transgenic plants that are from a single GFP−positive shoot have the same genotype. On the other hand, shoot regeneration may be inhibited by antibiotics during shoot culture. In the present study, for PS76 and H19, kanamycin alone seemed to be an effective selective agent for shoot regeneration and rooting. Our results indicated that 100 mg L^−1^ of kanamycin in the shoot culture medium was able to inhibit 96% of the initiating or greening of shoots (Figure 4). Moreover, we found that some leaves of regenerated plantlets were killed by kanamycin while other were not affected, which suggested that kanamycin can effectively reduce the chimera in the transformants. Roots were more sensitive to kanamycin (Figure 4H); we found that 50 mg L^−1^ of kanamycin in the rooting culture medium was effective to select against false positives. Based on the RNAi−APRR2 transformation (Figure 5) and earlier studies (e.g., [85]), cucumber seems to be very sensitive to hygromycin, which may need more optimization in the system. For these reasons (at least for PS76 and 9930), we suggest the use of 100 mg L^−1^ of kanamycin for the selection of cucumber transformants during shoot regeneration. When other genotypes are used, some preliminary work may be helpful to optimize the antibiotic concentrations. If the construct carries the GFP reporter gene, both screening systems could be considered to improve the selection efficiency. 

Many other measures have been tested to enhance *Agrobacterium* infection of cucumber explants including vacuum infiltration, sonication, micro−brushing of explants, or addition of antioxidant chemicals in the culture medium to reduce oxidative stresses caused by mechanical wounding (e.g., [28,29,30]). We used a vacuum pump to facilitate vacuum infiltration in this study but did not optimize this system (pressure and duration) or evaluate its efficiency. Thus, the transformation protocol we developed from this study could be used as the baseline for future improvement by testing and optimization of these methods.

### 4.5. T−DNA Insertion as a Source of Mutagenesis in Cucumber

During transformation, the T−DNA is integrated into the recipient genome randomly via illegitimate recombination [39]. This T−DNA insertion mutagenesis may result in the disruption of the functions of genes in which the T−DNA is inserted. Tens of thousands of mutant lines have been developed with this strategy in Arabidopsis and rice, representing a powerful tool in functional genomics studies (e.g., 92; [36,37,38,86,87]). Work on T−DNA insertion mutagenesis in cucumber is rare [16], which is probably due to the low transformation efficiency in cucumber, thereby producing very few transgenic plants and thus lacking the necessary population size to detect visibly recognizable mutant phenotypes. Since T−DNA mutagenesis is not the goal for most transformation projects, a lack of attention to those rare events is also possible. In this study, we produced many T_1_ seeds from T_0_ LL−OE transgenic plants, which allowed us to grow a large number of T_1_ plants in the field to identify mutants that were easy to recognize visibly (Figure 8 and Appendix A). Among a dozen mutants we identified, four were confirmed to be inheritable (Figure 8), while others still require additional investigation to confirm (Appendix A). These mutants could also be due to mutagenesis during tissue culture or spontaneous mutations, but the chance seems to be very low. An indirect piece of evidence is that all the mutants we identified were from LL−OE progeny. Most of those mutant plants also showed the expected phenotypes overexpressing the *LL* gene. Nevertheless, further validation is needed. A simple whole−genome resequencing or map−based cloning of these mutant alleles may confirm this. The scope of T−DNA insertion mutagenesis that does not have visible phenotypic changes in the LL−OE or other transgenic plants may also merit an evaluation. Regardless, work from this study suggested that *Agrobacterium*−mediated transformation is a potential source of T−DNA insertion mutagenesis. 

## 5. Conclusions

In general, North China−type cucumbers have a relatively higher shoot−regeneration rate and transformation efficiency than the US slicing or pickling cucumbers. Both Poinsett 76 and 9930 are genotypes of choice in cucumber genetic transformation studies. When working with either genotype, the following parameters could be considered: use *Agrobacterium* strain AGL1; harvest *Agrobacterium* at OD = 0.7; germinate seeds at 28 °C for 3 d for explant harvest; co−cultivate explants and bacterium at 23 °C for 2 d; select only yellowish explants for shoot culture; and use kanamycin as a selective agent at 100 mg L^−1^ for shoot regeneration and 50 mg L^−1^ for rooting. This protocol can be used as the baseline for further improvements to increase the transformation efficiency. In addition, *Agrobacterium*−mediated transformation is a good source of mutations from T−DNA mutagenesis. 

## Figures and Tables

**Figure 1 genes-14-00601-f001:**
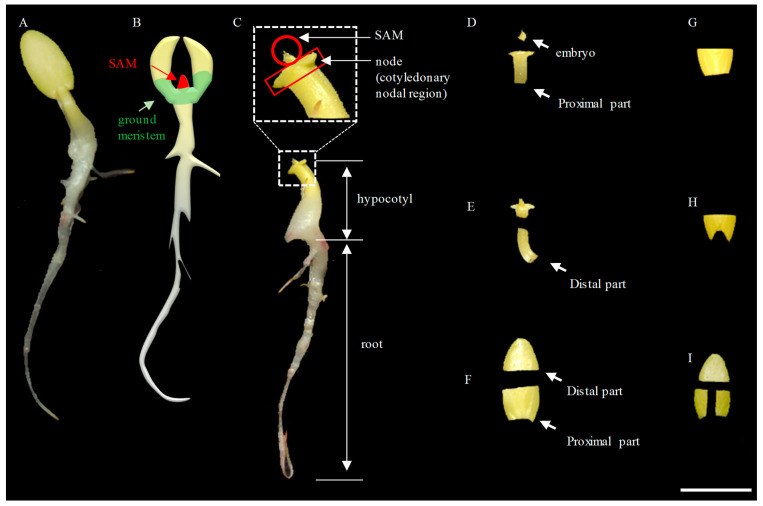
Harvest of different explants from 3−day−old cucumber seedlings for tissue culture. (**A**) An etiolated seedling cultivated in the dark at 28 °C at 3 d (72 h) after germination. (**B**) The positions of the shoot apical meristem (SAM) and ground meristem region on the seedling are highlighted in red and light green, respectively. (**C**) The different organs are shown. (**D**) Dissected SAM and proximal part of the hypocotyl. (**E**) Dissected SAM and distal part of the hypocotyl. (**F**) Cotyledons were cut in half transversely with distal and proximal segments. (**G**) The proximal part of a cotyledon was excised with a straight cut. (**H**) A V−shaped cut from the proximal end of the cotyledon. (**I**) The proximal parts of the cotyledon were cut into 2 pieces longitudinally. Scale bar = 1.0 cm.

**Figure 2 genes-14-00601-f002:**
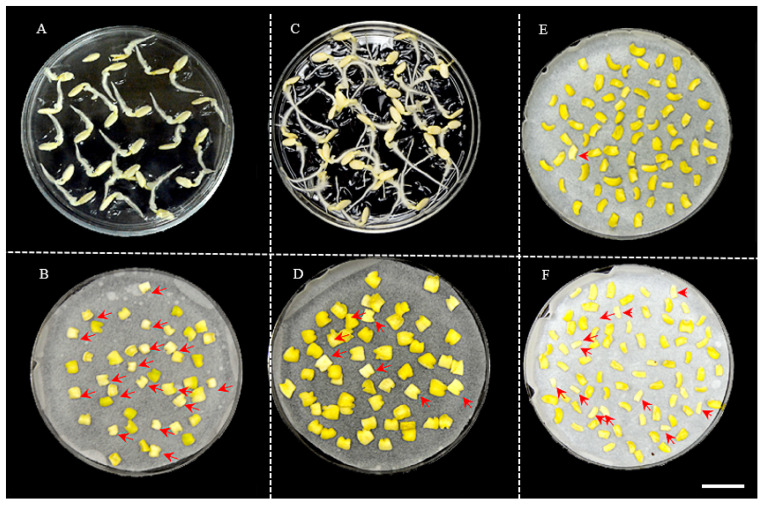
Frequency of whitish and yellowish explants was related with duration of seed germination and co−cultivation of explants. (**A**). Seeds germinated at 28 °C for 2 d. (**B**). Explants from 2−day germination were co−cultured in the dark for 2 d. (**C**). Seeds germinated at 28 °C for 3 d. (**D**). Explants from 3−day germination were co−cultured in the dark for 2 d. (**E**). Explants without inoculation after co−culture for 2 d. (**F**). Explants inoculated with *Agrobacterium* and co−cultured for 2 d. The red arrows indicate whitish explants that had poor shoot−regeneration ability. Scale bar = 1.0 cm.

**Figure 3 genes-14-00601-f003:**
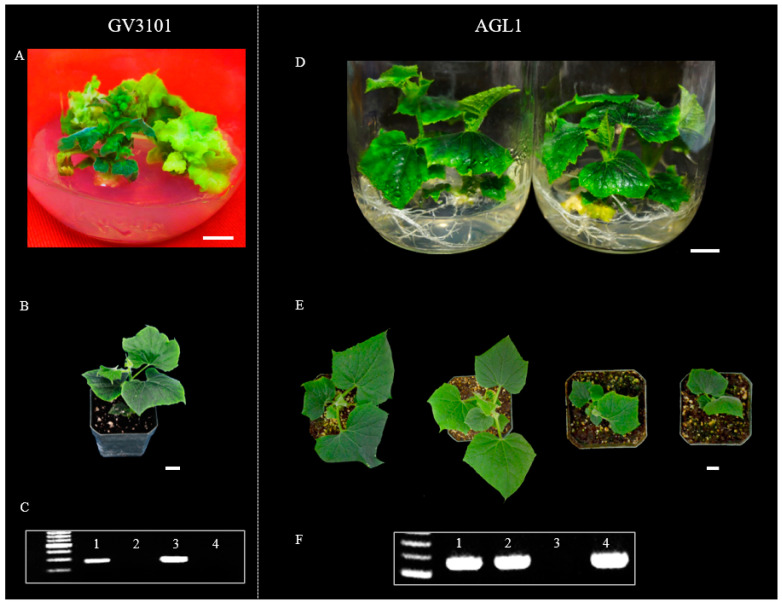
*Agrobacterium* stain AGL1 exhibited higher transformation efficiency than GV3101. Transgenic plants were developed with RNAi constructs for the *CsAPRR2* gene in a PS76 background. Plantlets in the jars were generated from GV3101−mediated (**A**) and AGL1−mediated (**D**) transformation, respectively. T_0_ transgenic plants grown in soil were mediated by GV3101 (**B**) and AGL1 (**E**), respectively. PCR verification identified positive T_0_ transgenic plants from GV3101−mediated (**C**) and AGL1−mediated (**F**) transformation, respectively. In (**C**,**F**), the first lane is a size marker, and lanes 1–4 are the positive control and three regenerated plantlets from independent transformation events. Scale bar = 1.0 cm.

**Figure 4 genes-14-00601-f004:**
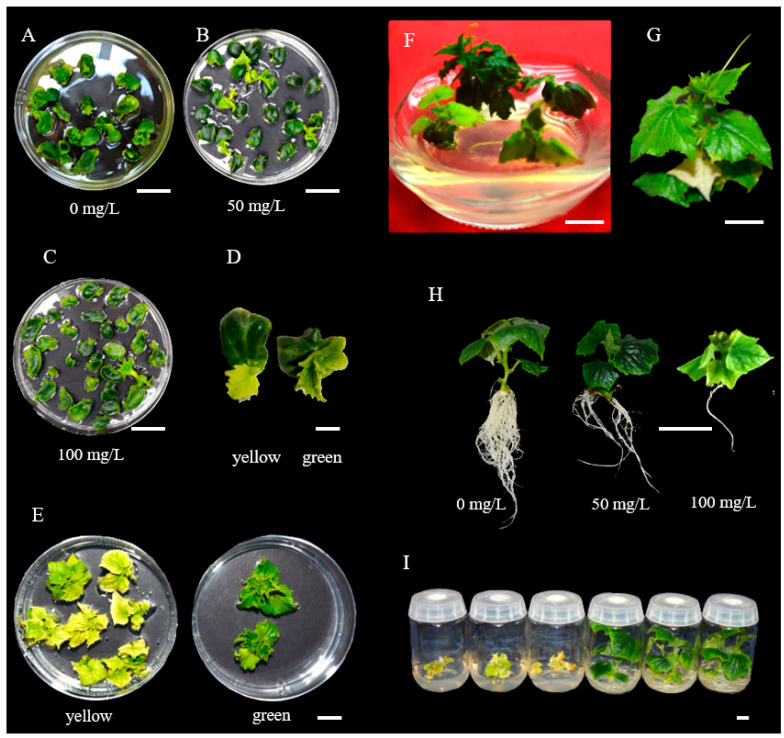
Antibiotic selection at different stages. (A–**C**) Explants in MS medium with 0, 50 and 100 mg L^−1^ of kanamycin. (**D**) Explants initiating yellow and green shoots. (**E**) Yellow and green plantlets growing in Petri dishes. (**F**) Plantlets growing in jars. (**G**) One albino true leaf in the plantlet was killed by kanamycin, which probably was due to the generation of a chimeric plantlet during subculturing. (**H**) T_0_ plantlets grown in rooting medium with different concentrations of kanamycin. (**I**) Plantlets in rooting MS medium. Scale bar = 1.0 cm.

**Figure 5 genes-14-00601-f005:**
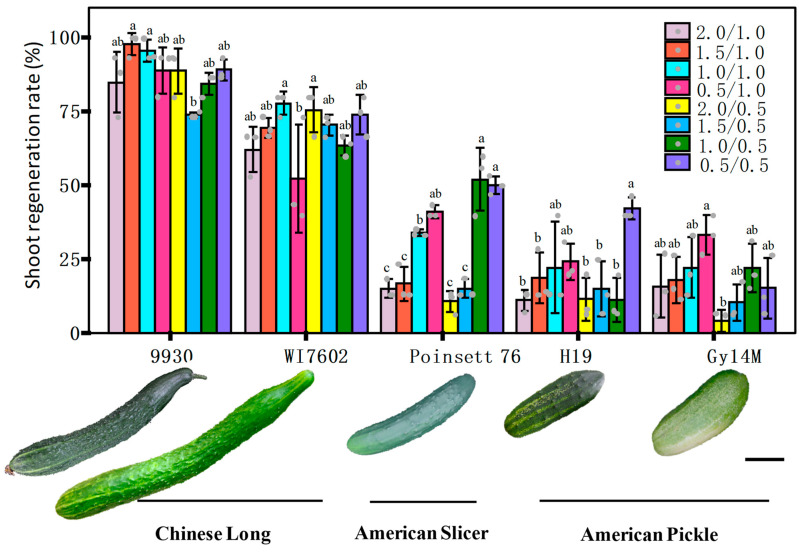
Effect of 6−BA and ABA phytohormones on shoot−regeneration rate (%) among five cucumber genotypes. There were eight combinations of 6−BA (at 0.5, 1.0, 1.5, and 2.0 mg L^−1^) and ABA (at 0.5 and 1.0 mg L^−1^). Fruit appearances of five genotypes and their market groups are shown under the bar graph. For each bar of the graph, the error bar is the mean ± SD (standard deviation) from three biological repeats. The letters a–c indicate statistically significant differences between means of the shoot−regeneration rate based on one−way ANOVA with the Tukey–Kramer test (*p* < 0.05). Scale bar = 5 cm. We compared the transformation efficiency between PS76 and H19 by developing transgenic plants overexpressing the *LL* gene driven by the CaMV 35S promoter ([25]). For H19 (Figure 6), 400 cotyledon explants were co−cultivated from 200 seeds with the *Agrobacterium* strain AGL1. After shoot regeneration and rooting, two plantlets were able to survive from culture in MS medium containing 100 mg L^−1^ of kanamycin, one of which was validated by PCR (Figure 6A–C). Consistent with the functions of the *LL* gene ([25]), the LL-OE transgenic plants displayed significantly large organ sizes for male and female flowers, leaves, and fruits (Figure 6B,D–G). Thus, in H19 LL−OE, the overall transformation efficiency was 0.25% (=1/400 × 100 for explants) or 0.50% (based on the number of seeds). For PS76, we germinated 2103 seeds and obtained 4925 explants from cotyledons and hypocotyls. Using the same transformation procedures as for H19, we eventually obtained 196 kanamycin−resistant plantlets. All the 196 putative transgenic plants were subjected to PCR validation, of which 24 were PCR−positive (TE = 0.49% with explants or 1.14% with seeds); 21 of the 24 positive plants displayed expected phenotypes ([25]).

**Figure 6 genes-14-00601-f006:**
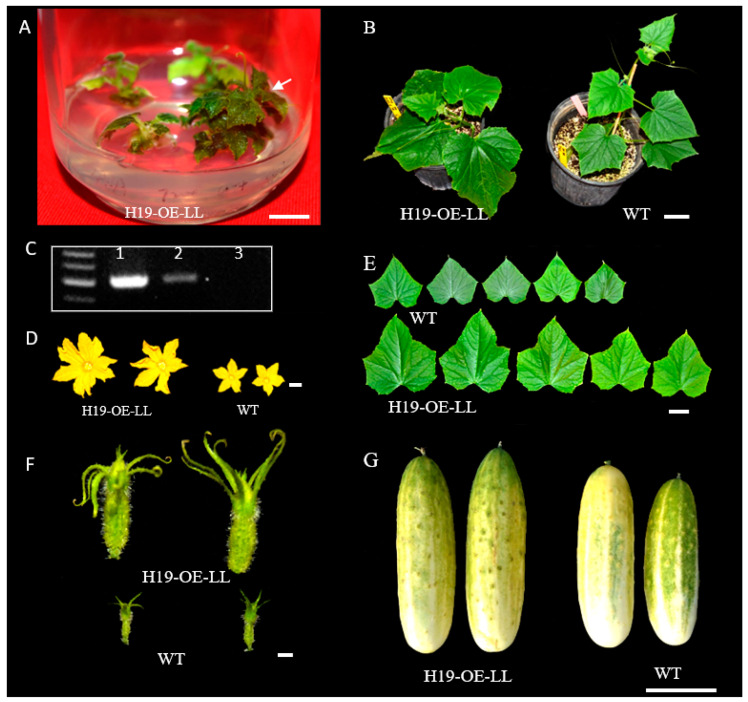
Development of transgenic plants overexpressing (OE) the *LittleLeaf* (*LL)* gene in pickling cucumber inbred line H19. (**A**) Regenerated plantlets in selective MS medium in a jar. (**B**) The regenerated plantlet (H19−OE−LL) was transplanted into soil together with the control (H19) plant. (**C**) One transgenic plant (lane #2) was validated by PCR. Lane 1 is the positive control, and the plant in lane 3 was a false positive. Comparison of the male (**D**) and female (**F**) flowers, first 5 true leaves (**E**), and mature fruits between the OE line and WT also supported that H19−OE−LL is a true transgenic plant overexpressing the *LL* gene with phenotypic effects. All plants shown are at T_0_. Scale bar = 1.0 cm in (A,D,F) and 5.0 cm in (**B**,**E**,**G**).

**Figure 7 genes-14-00601-f007:**
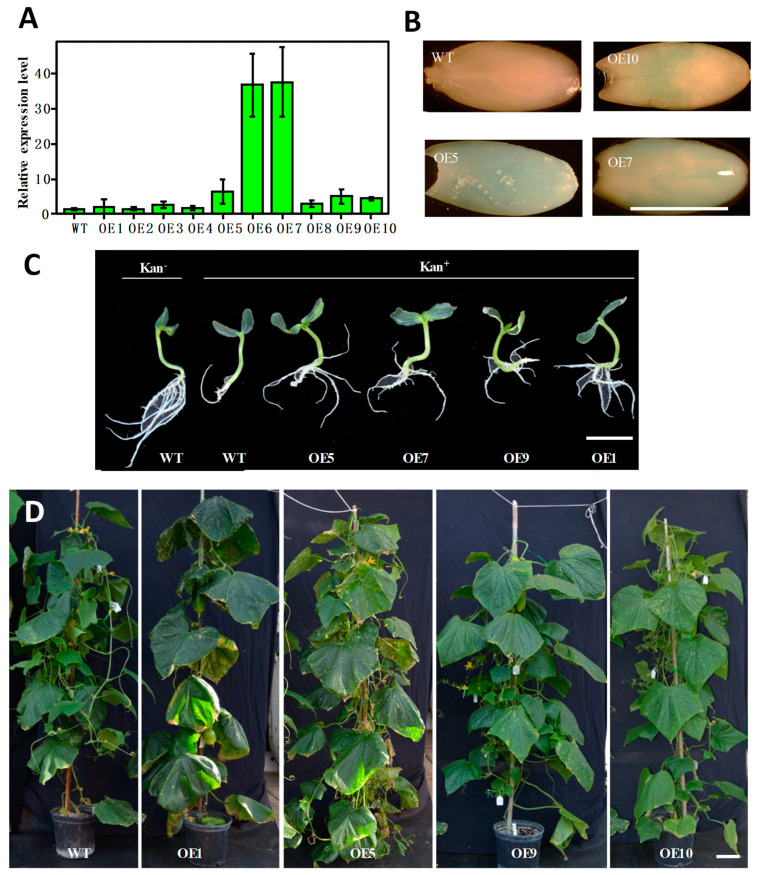
Validation and characterization of T_1_ transgenic plants overexpressing *LL* gene (OE) in PS76 background. (**A**) Examination of the expression level of *LL* in OE plants with qPCR. (**B**) Examination of *LL* expression in immature seeds with GUS staining. (**C**) Evaluation of kanamycin resistance of OE plants in MS medium with (Kan^+^) and without (Kan^−^) the selective agent. (**D**) OE transgenic plants had a larger leaf area than the WT. Scale bar = 1.0 cm (**A**–**C**) or 10.0 cm (**D**).

**Figure 8 genes-14-00601-f008:**
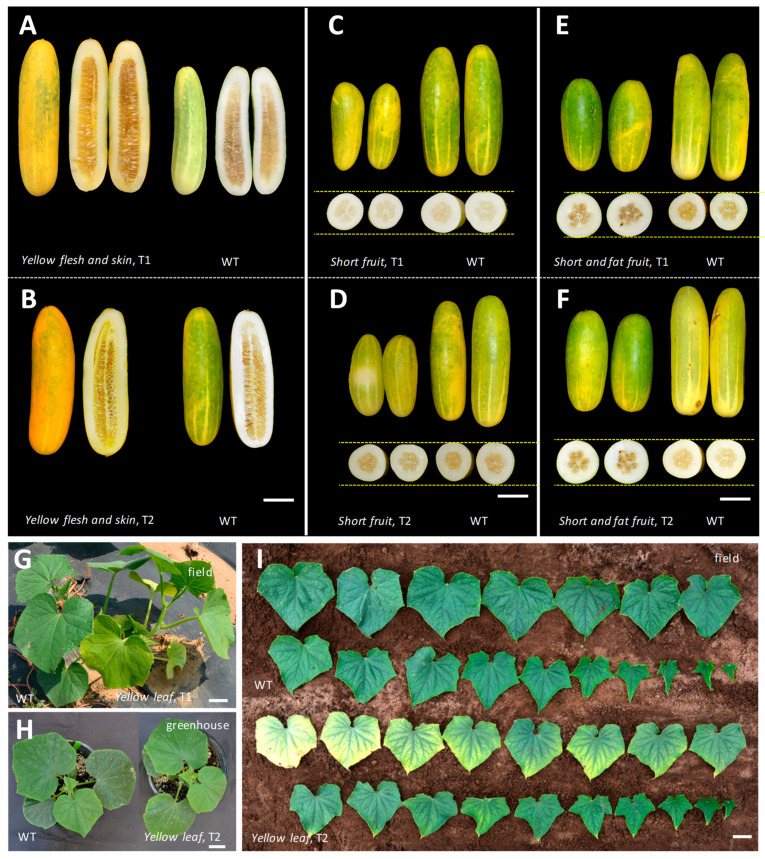
Examples of four leaf and fruit size or color mutants identified from *Agrobacterium*−mediated transgenic plants overexpressing the *LL* gene in the PS76 genetic background. The inheritability of each mutant phenotype was validated by consistent performance in both T_1_ and T_2_ generations under greenhouse and/or field conditions. The four mutants included *yellow flesh and skin* (**A**,**B**), *short and slim fruit* (**C**,**D**) *short and fat fruit* (**E**,**F**), and *yellow leaf* (**G**−**I**). In (**I**), for both the WT and the mutant, the leaves are in order from the oldest at the upper left to the youngest at the lower right. Phenotypes of each mutant in T_1_ and T_2_ generations are shown. Scale bar = 5.0 cm.

## Data Availability

All data pertinent to the reported work have been provided in the manuscript or in the Appendix A.

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
