# Peer review of "Improving Agrobacterium tumefaciens−Mediated Genetic Transformation for Gene Function Studies and Mutagenesis in Cucumber (Cucumis sativus L.)"

_genes, 2023, doi:10.3390/genes14030601_

Round 1

Reviewer 1 Report

Five monoecious cucumber varieties were used in the study. Transformation efficiency and strain effects were determined using two different methods. However, it is not clear whether these two methods were used in the five varieties. The aim of the study is to determine an effective transformation by testing some different factors and to determine the mutant potential of T-DNA insertion. In the study, many evaluations were made using different combinations of the protocol. At the same time, the difference in transformation efficiency rate between American and Chinese cultivars is clearly indicated. It is very useful to give the protocol in detail in the supplementary materials. In addition, the demonstration of phenotypic differences in both greenhouse and field conditions was illuminating. As a result of this study, the production method and duration of many transplant plants will provide convenience and save time for other researchers. However, it is not clearly stated in the article whether overexpression and knockdown or even two Agrobacterium strains are used in all cultivars. If used, there is a lack of transfer in statements and results. If not used, it should be explained why it was not used.
  P1L42. There are two Pan et al. 20202 references. Although these two references are mentioned separately in the references, it should be stated here which one they are referring to.
P3L40. The link is not working. P9L5. Why is T0 written in red? P9L19. 100 is written twice. P9L20. Were different numbers of plants used in these three different media?   P14L12. Should we consider only the 3 genes studied in the expression here, or do we think the TE ratios as a result of the overall study? Unfortunately, the expression is not clear. If we need to think about these 3 genes, why are the results not reported for PS76 and H19 as well? Fig6. A, B, and C should be written in parentheses. Fig7(d). There are two OE1s. What is the difference between them? FigS1. Please include explanations of all abbreviations.
Supp file S1. What do MGM and LB mean? Before using the abbreviation, make sure to use the full name first. Rajagopalan and Perl-Treves (2005), Feng et al. 2021b, Li et al. 2021a and Zhang et al. 2017 are not available in the reference. Sources are in references that are not included in the article: Bakhsh et al 2014, Ding et al 2015, Fengb et al 2021, Ikeuchi et al 2019, Koncz et al 1992, Li et al 2009, Liu et al 2016b, Lv et al 2020, Miao et al 2009, Nanasato and Tabei 2020, Prem and Rafael 2005, Tuantuan et al 2021, and Wang et al 2013.

Reviewer 2 Report

Cucumber transformation efficiency is low. In this study, evaluated the effects of several key factors affecting shoot regeneration rate and overall transformation efficiency in cucumber. The parameters (genotypes, Agrobacterium strain, kanamysin doses, hormone applications, etc.) giving the best results in genetic transformation studies for cucumber were determined. The Agrobacterium-mediated transformation protocol from this study could be used as the baseline for further improvement of cucumber transformation. This study, in which the most appropriate protocol is determined, is very important for the scientific world. It will shed light on gene editing studies in particular. I believe that the article will be accepted after the minor corrections mentioned below are made.

1. I suggest adding the author of the latin name in the title (Cucumis sativus L.)

2.The source spellings are not correct. It should be prepared in journal format

3.For each genotype, I propose describing the hormone doses that increase shoot regeneration.

4.Figure 5 shows that the standard deviation/standard errors are very high in some groups. It would be good to explain why. Has nonparametric statistical analysis been performed? For example, in the Gy14m 2.0/0.5 application, the values seem to be quite different. The reason for this wide variation in measurements can be interpreted.

5. In the Material Method section, it should be written how and in what way the statistical analyzes were made, and the number of repetitions should be stated.

6.It may be added how the 6-BA and ABA concentrations are determined. If there is a reference, it can be added.

7. Unit spellings should be the same:

1st page 31st line (mg/L)

5st page 1st line (mg L-1) etc.

8.Some corrections need to be made about reference writings:

a- While "Ulker et al" was written in the article, "Ülker" was written in the references list. It should be written in one type.

b- Again, while "OMalley" is written in the text, "O'Malley" is written in the reference list.

c-25. page 50 line should be N. Tabacum.(false)....tabacum (true).

d-24. page 4 line latina name should be italicized

e-24. page 9 line date is written in wrong place.

f.1. page 42 line "a" or "b" should be added to the "Pan et al.," reference

etc.

9. Abbreviations should be written in their long form in the first place. For example, although the MS description was first mentioned on page 2, line 29, the first long version was written on page 4, line 23.

10.In the primer sequences on 5st page 12st line, 3'….. 5' expression should be added

11.Some other typos should be corrected. For example, 9st page 19st line, written "100" twice.

Reviewer 3 Report

The abstract is sufficiently informative even when is read in isolation.

The introduction is informative to current investigations worldwide.

Materials and Methods

Page 3 Row 36-37 - It should be mentioned who supplies the seeds from cucumber lines.

Page 5 Row 32 – More details on how is measure size of roots, stem, flower and fruits?

Results

Page 11 – Row 31-32 “Overall, the shoot regeneration ability was significantly higher in 9930 and WI7402 than in PS76, H19 or Gy14M.” If we talk about significance, some significance test should be performed like T or F test.

Page 12 - Figure 5 – It should be mentioned the deviations in bars – what does it expressed? It can be range, standard deviation, standard error of mean, confidence intervals or others.

References

“Rajagopalan and Perl-Treves (2005)” is omitted.
